# Inhibition of the Hepatic Uptake of ^99m^Tc-Tetrofosmin Using an Organic Cation Transporter Blocker

**DOI:** 10.3390/pharmaceutics13071073

**Published:** 2021-07-13

**Authors:** Kodai Nishi, Masato Kobayashi, Minori Kikuchi, Asuka Mizutani, Yuka Muranaka, Ikumi Tamai, Keiichi Kawai, Takashi Kudo

**Affiliations:** 1Department of Radioisotope Medicine, Atomic Bomb Disease Institute, Nagasaki University, 1-12-4 Sakamoto, Nagasaki 852-8523, Japan; tkudo123@nagasaki-u.ac.jp; 2Faculty of Health Sciences, College of Medical, Pharmaceutical and Health Sciences, Kanazawa University, 5-11-80 Kodatsuno, Kanazawa 920-1192, Japan; kobayasi@mhs.mp.kanazawa-u.ac.jp (M.K.); mizutani.a@staff.kanazawa-u.ac.jp (A.M.); yukarisa93@stu.kanazawa-u.ac.jp (Y.M.); kei@mhs.mp.kanazawa-u.ac.jp (K.K.); 3School of Medicine, Nagasaki University, 1-12-4 Sakamoto, Nagasaki 852-8523, Japan; minolith04@gmail.com; 4Faculty of Pharmaceutical Sciences, Institute of Medical, Pharmaceutical and Health Sciences, Kanazawa University, Kakuma, Kanazawa 920-1192, Japan; tamai@p.kanazawa-u.ac.jp; 5Biomedical Imaging Research Center, University of Fukui, 23-3 Matsuokashimoaizuki, Eiheiji, Fukui 910-1193, Japan

**Keywords:** ^99m^Tc-tetrofosmin, small-animal SPECT/CT, hepatic uptake, transport mechanism, OCT1

## Abstract

The accumulation of high levels of ^99m^Tc-tetrofosmin (^99m^Tc-TF) in the hepatobiliary system can lead to imaging artifacts and interference with diagnosis. The present study investigated the transport mechanisms of ^99m^Tc-TF and attempted to apply competitive inhibition using a specific inhibitor to reduce ^99m^Tc-TF hepatic accumulation. In this in vitro study, ^99m^Tc-TF was incubated in HEK293 cells expressing human organic anion transporting polypeptide 1B1 (OATP1B1), OATP1B3, OATP2B1, organic anion transporter 2 (OAT2), organic cation transporter 1 (OCT1), OCT2, and Na^+^-taurocholate cotransporting polypeptide with or without each specific inhibitor to evaluate the contribution of each transporter to ^99m^Tc-TF transportation. In vivo studies, dynamic planar imaging, and single photon emission computed tomography (SPECT) experiments with rats were performed to observe alterations to ^99m^Tc-TF pharmacokinetics using cimetidine (CMT) as an OCT1 inhibitor. Time–activity curves in the liver and heart were acquired from dynamic data, and the ^99m^Tc-TF uptake ratio was calculated from SPECT. From the in vitro study, ^99m^Tc-TF was found to be transported by OCT1 and OCT2. When CMT-preloaded rats and control rats were compared, the hepatic accumulation of the ^99m^Tc-TF was reduced, and the time to peak heart count shifted to an earlier stage. The hepatic accumulation of ^99m^Tc-TF was markedly suppressed, and the heart-to-liver ratio increased 1.6-fold. The pharmacokinetics of ^99m^Tc-TF were greatly changed by OCT1 inhibitor. Even in humans, the administration of OCT1 inhibitor before cardiac SPECT examination may reduce ^99m^Tc-TF hepatic accumulation and contribute to the suppression of artifacts and the improvement of SPECT image quality.

## 1. Introduction

Coronary artery diseases are the leading cause of death in many developed countries. To decrease mortality and morbidity, signs need to be identified as early as possible. Single photon emission computed tomography (SPECT) is a useful diagnostic modality for cardiac disease because SPECT images can reflect cardiac function.

Radiotracers for cardiac SPECT, such as ^201^TlCl, ^99m^Tc-2-methyoxyisobutylisonitrile (^99m^Tc-MIBI), ^123^I-beta-methyl iodophenyl-pentadecanoic acid (^123^I-BMIPP), and 3-^123^I-metaiodobenzylguanidine (^123^I-MIBG), are in widespread clinical use [1,2,3]. In particular, ^99m^Tc-tetrofosmin (^99m^Tc-TF) is a well-known radiotracer for the detection of myocardial perfusion abnormalities and can quantify some cardiac functions [4].

Generally, radiotracer accumulation in a specific target organ is desirable, but most radiotracers also accumulate in other organs. In fact, ^99m^Tc-TF is known to accumulate in the hepatobiliary system depending on the dose of ^99m^Tc-TF injected, blood flow to the liver, and excretions from the liver [5]. High accumulations in the hepatobiliary system can cause imaging artifacts and interfere with diagnosis. Therefore, despite the application of various ingenious methods in the clinical stage to avoid adverse impacts, more effective methods are still required.

It is known that ^99m^Tc-TF is carried into cells by passive diffusion and membrane potential difference [6,7]. However, we speculated that drug transporters are also involved in the hepatic uptake of ^99m^Tc-TF because some ^99m^Tc-labeled compounds are transported via solute carrier (SLC) transporters and adenosine triphosphate-binding cassette (ABC) transporters in the liver [8]. In particular, P-glycoprotein (MDR1) and multidrug resistance-associated protein (MRP)1, MRP2, and MRP3 in ABC transporters are involved in the transport of ^99m^Tc-TF [9]. Previous studies have shown that ^99m^Tc-TF has an affinity for ABC transporters in the efflux of ^99m^Tc-TF into bile [9], but no reports appear to have clarified SLC transporters for the hepatic uptake of ^99m^Tc-TF. If such SLC transporters could be identified, the hepatic accumulation of ^99m^Tc-TF could be expected to be dramatically reduced when SLC transporters were inhibited by the competitive inhibition of SLC transporters. The aim of the present study was to improve the image quality of cardiac perfusion SPECT by reducing the liver uptake of ^99m^Tc-TF.

## 2. Materials and Methods

### 2.1. Uptake Experiments with HEK293 Cells

Transporter affinity was measured using human embryonic kidney 293 (HEK293) cells transiently expressing human organic anion transporting polypeptide 1B1 (OATP1B1), OATP1B3, OATP2B1, organic anion transporter 2 (OAT2), organic cation transporter 1 (OCT1), OCT2, Na^+^-taurocholate cotransporting polypeptide (NTCP), and mock cells. All cells were purchased from GenoMembrane Inc. (Kanagawa, Japan). Dulbecco’s modified Eagle’s medium (DMEM) and fetal bovine serum (FBS) were purchased from Wako (Osaka, Japan) and Thermo Fisher Scientific K.K. (Yokohama, Japan), respectively. A ^99^Mo-^99m^Tc generator (925 MBq) and tetrofosmin kit were purchased from FUJIFILM Toyama Chemical Co. (Tokyo, Japan) and Nihon Medi-Physics Co. (Chiba, Japan), respectively.

For the competitive inhibition assay, bromosulflein (BSP, a substrate specific to OATP), p-aminohippuric acid (PAH, a substrate specific to OAT), cimetidine (CMT, a substrate specific to OCT), and taurocholate hydrate (TC, a substrate specific to NTCP) were purchased from Sigma Chemical (St. Louis, MO, USA).

The cells were cultured in DMEM supplemented with 10% (*v*/*v*) FBS, 100 U/mL penicillin and 100 µg/mL streptomycin at 37 °C with 5% CO_2_. The uptake experiment with HEK293 was performed according to the methods described by Kobayashi et al. [8,9].

Two days before the uptake experiment, HEK293 cells were prepared at 4 × 10^5^ cells/well in 12-well plastic plates. Cells were pre-incubated for 5 min using modified Hank’s balanced salt solution (MHBS). After pre-incubation, cells were then incubated with 0.5 mL of MHBS containing ^99m^Tc-TF (150 kBq/well) for 5 min as a control condition. In the competitive inhibition experiment, cells were incubated with ^99m^Tc-TF and 1.0 mM of inhibitors for 5 min under the same control condition. At the end of incubation, each well was rapidly washed with ice-cold MHBS. Cells were solubilized in 0.1 mL of 0.1 N NaOH and mixed with an ASC-II Scintillation Cocktail (GE Healthcare UK, Little Chalfont, UK). Radioactivity was measured using a liquid scintillation counter (LSC-5100; Hitachi, Tokyo, Japan). The amount of protein in cells was quantified using the bicinchoninic acid method [10].

### 2.2. Dynamic Planar and SPECT Imaging

A total of 6 male Wistar rats (8 weeks old; body weight, 200–230 g) were purchased from Japan SLC, Inc. (Shizuoka, Japan). Animal studies were performed in accordance with the recommendations of the Fundamental Guidelines for Proper Conduct of Animal Experiment and Related Activities in Academic Research Institution under the jurisdiction of the Ministry of Education, Culture, Sport, Science, and Technology. The Animal Care and Use Committee of Nagasaki University approved all of the experimental protocols (approval number: 1506171238-2, 17 June 2015). Before the experiments, the rats were fasted for about 4 h.

All imaging studies were performed using Triumph combined PET/SPECT/CT systems (TriFoil Imaging, Chatsworth, CA, USA). Control rats were intravenously administered 500 µL of saline (Otsuka Pharmaceutical Co., Tokyo, Japan), and treated rats were intravenously administered CMT (200 µg/500 µL) as the OCT1 inhibitor. Six minutes later, ^99m^Tc-TF (80 MBq/500 µL/60 s) was injected via the tail vein. Rats were anesthetized with 1.5% isoflurane, and dynamic planar imaging was performed for 4.5 min at 5 s/frame. After dynamic planar acquisition, SPECT was performed. SPECT acquisitions were performed for 21 min with 64 views over 360°, 20 s/projection, using a 60 mm radius of rotation, starting 6 min after ^99m^Tc-TF injection. After SPECT, CT was performed for anatomical reference.

### 2.3. Image Analysis

Dynamic planar images were analyzed using VivoQuant 2.3 (INVICRO, Boston, MA, USA). Regions of interest (ROI) were placed over the liver and heart. Time–activity curves were calculated from ROI data and corrected by injected dose.

SPECT data were reconstructed using a 3D-maximum-likelihood expectation maximization algorithm (50 iterations). CT and SPECT data were processed and analyzed using OsiriX MD (Pixmeo, Geneva, Switzerland). ROIs were placed over the liver and heart to obtain the total uptake of ^99m^Tc-TF. Heart-to-liver ratios were calculated.

### 2.4. Statistical Analysis

All results are given as the mean of least four experiments, expressed as mean ± SD. Data were analyzed using the *t*-test or paired *t*-test, and values of *p* < 0.05 were considered statistically significant. All statistical analyses were performed using Prism version 7.0 software (Graphpad Software Inc., San Diego, CA, USA).

## 3. Results

Table 1 and Figure 1 indicate ^99m^Tc-TF accumulation levels in HEK293 cells. No significant differences were apparent between the control group and the inhibition group in OATP1B1, OATP1B3, OATP2B1, OAT2, or NTCP. However, for OCT1 and OCT2, the accumulated amount of ^99m^Tc-TF was decreased by competitive inhibition (*p* < 0.05). Thus, OCT1 and OCT2 were speculated to be involved in the intracellular transportation of ^99m^Tc-TF.

OCT1 inhibition also affected the in vivo pharmacokinetics of ^99m^Tc-TF. As shown in Figure 2, the liver counts of ^99m^Tc-TF reached a steady state in about 1 min after injection. In control rats, steady-state values were higher than in CMT-treated rats. On the other hand, the heart counts of ^99m^Tc-TF were markedly lower than those in the liver, and no difference was identified between control rats and CMT-treated rats. However, in CMT-treated rats, the time to peak heart count tended to shift to an earlier stage.

Figure 3 shows SPECT images in maximum intensity projection (MIP) mode. The cardiac uptake levels of ^99m^Tc-TF did not differ between the control rats and the CMT-treated rats. However, the tendency toward hepatic uptake levels differed markedly. In the control rats, ^99m^Tc-TF showed accumulation at a high level throughout the whole liver. On the other hand, the hepatic uptake of ^99m^Tc-TF was suppressed by CMT preloading. The heart-to-liver ratio increased 1.6-fold by the administration of CMT (Figure 4).

## 4. Discussion

In the present study, two main points merit attention. The first point is that the transport mechanism of ^99m^Tc-TF has been elucidated. Liver accumulation can be suppressed by inhibiting the uptake of ^99m^Tc-TF via OCT1. The second point is that this method appears to offer a high possibility of clinical application, given that CMT is an inexpensive and familiar medicine that can be administered as an OCT1 inhibitor.

One of the advantages of cardiac SPECT is its capability to evaluate myocardial perfusion. The under- and overestimation of myocardial perfusion can lead to the misdiagnosis of heart diseases, especially for ischemic heart disease directly. Radiotracers for cardiac perfusion diagnosis using ^99m^Tc tend to accumulate in the liver. Excessive accumulation in the liver adversely affects the evaluation and diagnosis of the left ventricular inferior wall in contact with the liver. Methods to suppress accumulation in the liver are thus necessary. To maintain the diagnostic performance of cardiac SPECT, various methods have been reported. For example, Hara reported that the quality of cardiac SPECT images of the inferior wall improved by drinking soda water before SPECT with the expansion of the gastric wall and by avoiding the reflux of bile containing high radioactivity [11], while Cherng reported that the liver excretion of ^99m^Tc-TF is accelerated by drinking lemon juice [12]. Some research has been conducted to promote the excretion of radiotracers from the liver and to improve the image quality of cardiac SPECT by eating fatty meals, drinking milk, and so on [13]. However, those are not ideal methods, because there are drawbacks, such as susceptibility to individual differences and the difficulty of global standardization. By provoking the alteration of the pharmacokinetics of ^99m^Tc-TF with drugs, not with food or drink, dosage per body weight can be precisely specified along with the timing of administration. In addition, with these methods, the preparation of foods or drinks was based on the idea of promoting the excretion of ^99m^Tc-TF accumulation in the liver. However, the rate of excretion is susceptible to individual differences and physical conditions. Our research group therefore focused on the transporter mechanism in the liver and methods to block the hepatic uptake of ^99m^Tc-TF.

First, our research group attempted to identify the transporters that carry ^99m^Tc-TF into liver cells. From the in vitro results, OCT1 and OCT2 are revealed to be strongly involved in the intracellular transport of ^99m^Tc-TF. OCT1 is therefore known to be primarily expressed on the sinusoidal membrane of hepatocytes, and also on the basolateral membrane of tissues such as the heart, small intestinal enterocytes, renal proximal tubular cells, and so on [14,15]. On the other hand, OCT2 is not expressed in the liver [14,15,16]. We thus assumed that the uptake of ^99m^Tc-TF into hepatocytes would be more affected than that into the heart when OCT1 is inhibited.

CMT is known to show high affinity for OCT1 and OCT2 [17,18,19], is low in toxicity, and has a long history of clinical usage for the treatment of peptic ulcers. The CMT concentration was determined by reference to blood CMT level in humans and the difference in metabolic rates between humans and rats.

Dynamic planar imaging acquisition was conducted to track the pharmacokinetics of ^99m^Tc-TF at the early stage of administration. Despite the almost identical rates of increase in time–activity curves, the liver counts of the rats treated with CMT at a steady state were decreased by about 20% compared to those of the control rats. This is because the uptake of ^99m^Tc-TF from blood into the liver via OCT1 was inhibited by CMT. Accordingly, the amount of ^99m^Tc-TF trapped in the liver decreased, and the relative ^99m^Tc-TF inflow into the heart immediately increased after administration. For these reasons, the timing of peak heart count was inferred to shift to an earlier stage. In clinical practice, myocardial perfusion SPECT image acquisition using ^99m^Tc-TF is started 1 h after radiotracer administration in order to wait for the reduction in the liver accumulation level of ^99m^Tc-TF. On the other hand, Figure 2 indicated that the liver accumulation level of ^99m^Tc-TF was kept low by preloading with CMT. The present results suggest the possibility of reducing the time required for myocardial perfusion SPECT inspection using ^99m^Tc-TF.

The alteration of the ^99m^Tc-TF pharmacokinetics caused by CMT is also reflected in the SPECT images. The results of SPECT imaging show improvements in image quality, with obvious decreases in intrahepatic accumulation, and the clarification of the boundary between the inferior cardiac wall and the liver. Figure 4 indicates not only the uptake ratio of ^99m^Tc-TF, but also contrast between the heart and liver on SPECT imaging. The experimental results show that the heart–liver contrast was significantly improved by preloading with CMT. Of course, CMT is predicted to also inhibit OCT1 expressed in other organs. However, the effects of OCT1 inhibition on other organs are inferred to be negligible from the results of dynamic planar imaging and the SPECT imaging studies.

The expression level and affinity to the substrates of OCT1 differ between species. A difference in the transport activity of OCT1 is also seen between species. However, no marked difference in substrate selectivity has been reported by species [20,21]. According to Wang et al., OCT1 is expressed at 5-fold higher levels in the human liver as compared to the rat liver [22]. More drastic effects would presumably be obtained if our method were to be applied to humans.

In addition, the present study points out the important issue of drug combinations. As stated already, CMT is used as a medication for reducing acid levels in the stomach. Image quality would differ between patients already medicated with and without CMT. The interpretation of those images could thus be influenced by information on concomitant medications. Medical staff need to be aware of the influence of medications taken on a daily basis and need to explain such issues to patients when planning cardiac SPECT examinations.

## 5. Limitations

All findings obtained in the present study were based on the results of in vitro experiments and animal experiments. The relevance of these findings to humans needs to be validated with appropriate clinical research. In addition, there is no certainty that the method of suppressing liver accumulation by OCT1 inhibition can also applied to ^99m^Tc-MIBI.

## 6. Conclusions

In the liver, OCT1 is strongly involved in the intracellular transport of ^99m^Tc-TF. The administration of OCT1 inhibitors can reduce the hepatic accumulation of ^99m^Tc-TF and improve image quality for cardiac SPECT.

## Figures and Tables

**Figure 1 pharmaceutics-13-01073-f001:**
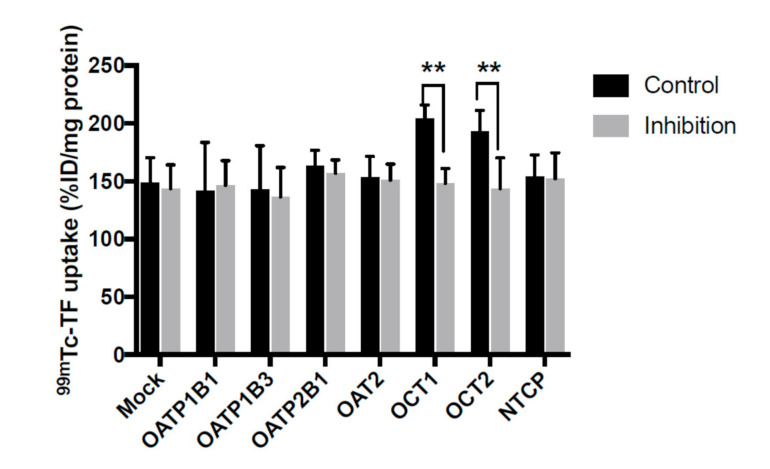
The uptake of ^99m^Tc-TF in human HEK 239 cells expressing SLC transporter. The uptake of ^99m^Tc-TF was significantly decreased by the inhibitors of OCT1 and OCT2. ** *p* < 0.01 vs. control condition, Student’s *t*-test.

**Figure 2 pharmaceutics-13-01073-f002:**
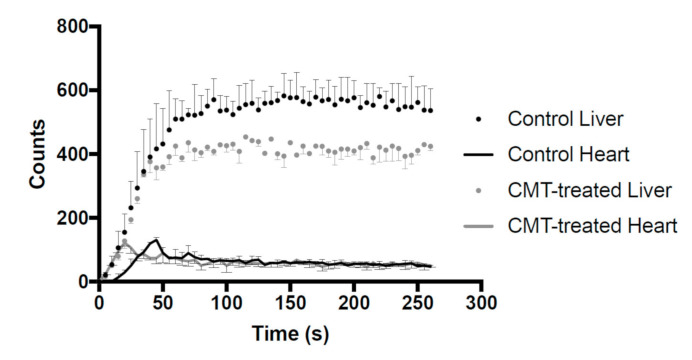
The time–activity curve of radioactivity in the liver and heart acquired by dynamic planar imaging. Data acquisition was performed for 4.5 min at 5 s/frame. Solid line: control rats. Dotted line: CMT-treated rats.

**Figure 3 pharmaceutics-13-01073-f003:**
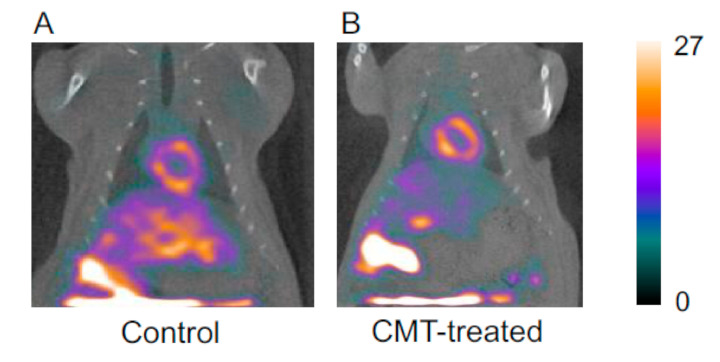
The SPECT images of a control rat (**A**) and a CMT-treated rat (**B**) injected with 80 MBq of ^99m^Tc-TF. The accumulation of ^99m^Tc-TF is high in the liver, but hepatic accumulation is markedly decreased in CMT-treated rat liver and the boundary between the heart and the liver is clear.

**Figure 4 pharmaceutics-13-01073-f004:**
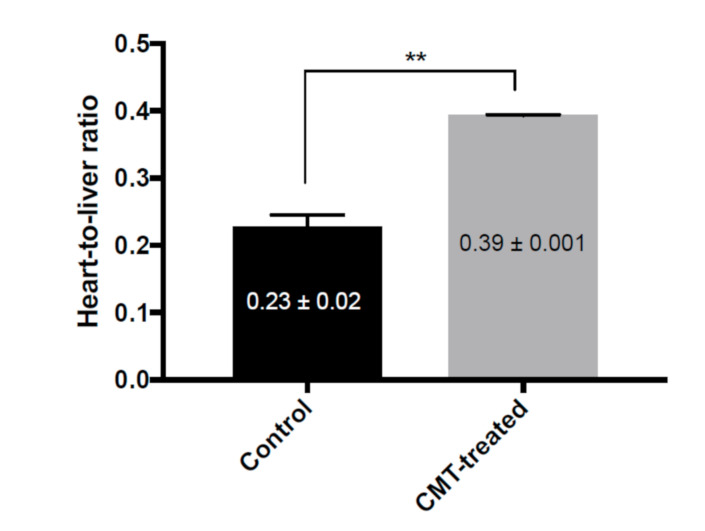
The ratio of ^99m^Tc-TF uptake in the heart and the liver. The heart-to-liver ratio is calculated as: (radioactive count for the whole heart)/(radioactive count for the whole liver). ** *p* < 0.01 vs. control, Student’s *t*-test.

**Table 1 pharmaceutics-13-01073-t001:** Details from Figure 1.

	Control (%ID/mg Protein)	Inhibition (%ID/mg Protein)
Mock	148.6 ± 22.0	143.6 ± 20.6
OATP1B1	141.5 ± 42.1	146.5 ± 21.3
OATP1B3	142.8 ± 38.0	136.5 ± 25.4
OATP2B1	163.7 ± 13.0	157.1 ± 11.4
OAT2	153.2 ± 18.3	150.9 ± 13.9
OCT1	204.3 ± 11.6	148.3 ± 12.8
OCT2	193.1 ± 18.2	143.7 ± 26.6
NTCP	153.9 ± 18.8	152.2 ± 22.4

## Data Availability

Not applicable.

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
