# Peer review of "Inhibition of the Hepatic Uptake of 99mTc-Tetrofosmin Using an Organic Cation Transporter Blocker"

_pharmaceutics, 2021, doi:10.3390/pharmaceutics13071073_

Round 1

Reviewer 1 Report

The authors investigated the transporters involved in hepatic uptake of TF in order to evaluate the hypothesis that inhibition of hepatic uptake could improve the quality of myocardial images.  Studies with transfected cell lines in vitro suggested that OCT1 might be involved and imaging studies in rats with or without CMT administration were able to demonstrate the effect of inhibition of OCT1 on liver accumulation and thus heart to liver ratios.

I have a number of minor questions and suggestions:

Title:  Would it read better as: “Inhibition of hepatic uptake of…”?

Introduction, lines 41-44. Wouldn’t it be better to use a more specific term than “cardiac diseases” in this paragraph?  References 1-4 only refer to myocardial perfusion imaging, so perhaps “coronary artery disease” should be used.  Although BMIPP and MIBG have metabolic and neurotransmitter mechanisms, the vast majority of nuclear cardiology involves myocardial perfusion.

Also, references 1, 2, and 4 are 20-30 years old.  Perhaps at least one should be replaced with a current reference, perhaps a practice guideline from SNM or EANM.

Re: Inhibition studies, lines 86-94.  Were the inhibitors added at the same time as the tracer, or beforehand as is often done in this sort of study?  Also, how was a uniform inhibitor concentration of 1 mM selected?  Isn’t the relative potency of inhibitors a factor?

Species: line 99 refers to Wistar rats.  Line 144 says mouse and mice.  Also lines 148-149.

Figure 2: It is difficult to differentiate the lines.

Peak heart counts, line 152.  I would like to see a little more evidence for the CMT-induced earlier peak counts.  Figure 2 shows an individual rat each for control and CMT.  If anything the control peak looks late: TF highly extracted on first pass.

Figure 4: Isn’t a heart to liver ratio usually calculated as average counts/pixel rather than counts/whole organ?

The references are not formatted consistently, e.g. capitalization, proper superscripts.

Reviewer 2 Report

The study by Nishi et al examined the uptake of 99mTc-tetrofosmin (TF) in the presence of organic cation transporter blocker cimetidine (CMT) in vitro and in vivo. The study is well written and presented clearly. CMT was shown to inhibit hepatic uptake of 99mTc-TF and may be potentially useful in the clinic.

There are a few points that I believe could improve the manuscript.

  1. Figure 1 demonstrates that transfection of HEK293 cells with OCT1 and OCT2 increases the uptake of 99mTc-TF compared to mock transfected cells. This is a useful finding that the authors may have missed. A comparison between all the transfected HEK293 groups sound be performed.
  2. Inclusion of additional cell lines such as HEPG2 and SK-HEP1. The authors discuss hepatobiliary clearance, the sinusoid and hepatocytes in the introduction and discussion sections. However, there is only examination of HEK293 cells in this study. HEPG2 cells should be examined to confirm the finding of OCT1 transporter inhibition decreases 99mTc-TF uptake in the liver. As this is the key finding and title of the paper additional in vitro work should be performed. Isolated rat hepatocytes could also be examined.  
  3. In the limitations section the authours highlight the method of suppressing liver accumulation by OCT1 inhibition is not applicable to 99mTc-MIBI in humans. Mouden et al (Nucl Med Commun. 2015 Feb;36(2):143-7) has also reported patients on 2 week PPI prior to receiving 99mTc-TF also have stomach wall uptake and artifact presence. This should be discussed and used to justify the use of iv and not oral administration of CMT in the current study.

Reviewer 3 Report

The introduction is well written, concise and clear.

Methods:

Authors could include a separate paragraph where the inhibitors are described (purcha information and function). So that the description of the uptake experiments is more linear.

Uptake experiments: was the incubation of 99mTc-TF and the inhibitors for 5 minutes? This should more clearly said.

Why wasn’t cimetidine tested in in vitro studies?

Results:

Why is the uptake of 99mTc-TF in control cells of OCT1 and 2 higher compared to other control cells? HAve the authors performed the in vitro studies in parallel with all inhibitors?                                                                                               

The authors should include the time-activity curves from all rats

Figure 2: the authors should use different lines for control liver and control heart. The same for CMT-liver and CMT-heart. It is otherwise very confusing

Line 174: do the authors can add a reference to support this statement?

Line 195: change in vivo with in vitro

Reviewer 4 Report

The paper of Kodai Nishi et al. is an experimental, interesting study with an work protocol presented in detail. Indeed, intrahepatic accumulation of 99mTc radiolabeled isonitriles may interfere with the outcome of myocardial scintigraphy. CMT administration could be easy and efficient, but without changing the image acquisition protocol.

A few things need to be improved in the article:

Q1. Please clearer explanation / reference (s) for the statement: "we consider that drug transporters are involved in the hepatic uptake of 99mTc-TF ..." (line 58)

Q2. Figure 2 need some improvement:

- it would be more appropriate to be represented the mean values (with SD) for all the subjects studied

- the correspondence between each line and its meaning is not  clearly specified (there are 2 lines with the same pattern with 2 different meanings...). Even if the explanation in the text helps, this aspect could be improved. Please correct.

- at the top of Figure 2 (and also for Figure 3) the name of the figure appears again - please standardize / correct this aspect.

Q3. Line 168: "two main points merit attention" - is not followed by the explanation of the respectives two points. Please correct.

Q4. The optimal time for acquisition of images in myocardial scintigraphy with 99mTc isonitriles (60 minutes, both for 99mTcMIBI and 99mTcTF, demonstrated to have similar kinetics) was chosen after in vitro radiotracer kinetics studies (examples of refferences below). As a result, the proposal, at line 214: „ ... possibility of reducing the time ... 99mTc-TF ...” cannot be acceptable in relation with the liver excretion.

Here are some refferences for more explanations:

Arbab AS, Koizumi K, Toyama K, Arai T, Araki T. Technetium-99m-tetrofosmin, technetium-99m-MIBI and thallium-201 uptake in rat myocardial cells. J Nucl Med. 1998 Feb;39(2):266-71. PMID: 9476934.

Maublant JC, Moins N, Gachon P, Renoux M, Zhang Z, Veyre A. Uptake of technetium-99m-teboroxime in cultured myocardial cells: comparison with thallium-201 and technetium-99m-sestamibi. J Nucl Med. 1993 Feb;34(2):255-9. PMID: 8429344.

Please modify.

Q5. The statement in line 244: „...the method of supressing liver accumulation by OCT1 inhibition is not applicable to 99mTc MIBI in humans” is not supported by either the data in the article or other scientific data.

Please correct.

Also, please check the uniformity of abbreviations (line 142 - Fig 2; line 153 - Figure 3; line 158 - Fig.4) and character size (line 266-268).

Round 2

Reviewer 2 Report

The authors have satisfactory addressed points 1 and 3. However, I do not believe they have addressed point 2 and additional experimentation is required.

If HEPG2 or SK-Hep1 are not applicable, then I suggest (as was suggested in the first review and not mentioned in your response) that isolated rat hepatocytes should be examined in vitro. Examining the uptake of  99mTc-TF in hepatocytes with and without the OCT1/2 inhibitors will determine if these are the only channels are involved in 99mTc-TF uptake. The scavenging liver sinusoidal endothelial cells should also be examined to determine which cells in the liver are taking up 99mTc-TF.    

Reviewer 3 Report

The authors addressed all my questions and the manuscript can be accepted for publication in my opinion.

Author Response

Thank you so much for your kind review comment.